# Prediction of Tinnitus Perception Based on Daily Life MHealth Data Using Country Origin and Season

**DOI:** 10.3390/jcm11154270

**Published:** 2022-07-22

**Authors:** Johannes Allgaier, Winfried Schlee, Thomas Probst, Rüdiger Pryss

**Affiliations:** 1Institute of Clinical Epidemiology and Biometry, University of Würzburg, 97070 Würzburg, Germany; ruediger.pryss@uni-wuerzburg.de; 2Department of Psychiatry and Psychotherapy, University of Regensburg, 93053 Regensburg, Germany; winfried.schlee@gmail.com; 3Department for Psychotherapy and Biopsychosocial Health, University for Continuing Education Krems, 3500 Krems, Austria; thomas.probst@donau-uni.ac.at

**Keywords:** tinnitus, gradient boosting machine, mobile health, machine learning, multimodal data, explainable machine learning

## Abstract

Tinnitus is an auditory phantom perception without external sound stimuli. This chronic perception can severely affect quality of life. Because tinnitus symptoms are highly heterogeneous, multimodal data analyses are increasingly used to gain new insights. MHealth data sources, with their particular focus on country- and season-specific differences, can provide a promising avenue for new insights. Therefore, we examined data from the TrackYourTinnitus (TYT) mHealth platform to create symptom profiles of TYT users. We used gradient boosting engines to classify momentary tinnitus and regress tinnitus loudness, using country of origin and season as features. At the daily assessment level, tinnitus loudness can be regressed with a mean absolute error rate of 7.9% points. In turn, momentary tinnitus can be classified with an F1 score of 93.79%. Both results indicate differences in the tinnitus of TYT users with respect to season and country of origin. The significance of the features was evaluated using statistical and explainable machine learning methods. It was further shown that tinnitus varies with temperature in certain countries. The results presented show that season and country of origin appear to be valuable features when combined with longitudinal mHealth data at the level of daily assessment.

## 1. Introduction

Subjective Tinnitus is an auditory phantom perception which is subjectively perceived by the tinnitus patient in the ears or head in the absence of a corresponding external sound. The prevalence of subjective tinnitus in western societies is estimated at 8–28% [1,2]; 1–4% of the population suffer severely from tinnitus and seek medical help because of, e.g., sleep disturbances, anxiety, or concentration problems [3,4,5]. If there is an identifiable cause for the tinnitus, it is called secondary tinnitus. Identifiable causes can be, for example, a vascular disease or perforated eardrum. Primary tinnitus is when there is no identifiable cause. The term “tinnitus disorder” has been recently defined to describe these severe condition where subjective tinnitus is associated with suffering of the affected people [2]. However, on an individual basis, the suffering of the tinnitus patients can be reduced, for example, by the use of a cognitive behavioral therapy [6].

Note that efforts are constantly made to learn more about the heterogeneity of symptom profiles of tinnitus patients. However, the required data sources are often missing. As the proliferation of smartphones has led to powerful mobile health solutions (denoted as mHealth solutions), which might fill the gap of promising data sources, this paper makes use of mHealth data for tinnitus symptom profile research. Therefore, we investigate data of users from an mHealth platform called TrackYourTinnitus (TYT). TYT users report tinnitus symptoms on a daily basis. The hypothesis of the investigation is to reveal differences between these users across countries and seasons. Interestingly, related research has only been presented on data that was gathered without any mHealth context. For example, a study on seasonal changes in tinnitus symptomatology concluded that searches for tinnitus aspects are higher in winter than in summer, as well as when comparing countries [7]. Another work suggests an association with depression, which constitutes a common comorbidity of tinnitus, and season [8]. More specifically, the same work provides Internet-based evidence for the epidemiology of seasonal depression. The results suggest that internet searches for depression by people at higher latitudes are more affected by seasonal changes, while this phenomenon is faded out in tropical areas. Moreover, more than 70 years ago it was clinically observed that tinnitus increases during winter months [9,10]. Seasonal affective disorders (SAD), in turn, were studied by the authors of [11]. They conclude that SAD are present if symptoms occur during winter months and entirely disappear in summer.

Thus far, differences in gender have been found to play a role in tinnitus research [12,13,14,15]. Women with tinnitus more frequently reported cardiovascular disease, fibromyalgia, or burnout, among others. In contrast, men with tinnitus were more frequently associated with alcohol use, Ménière’s disease, anxiety syndrome, and panic disorders. In addition, women tend to be more sensitive to environmental sounds, while men have greater difficulty following a conversation. Age is another variable that affects the loudness of tinnitus. As a rule, the older the patient, the more severe the tinnitus symptoms. Therefore, from the perspective of the domain, age is a suitable feature for machine learning (ML) algorithms [16]. Tinnitus can have a significant impact on everyday functioning [17]. For example, the Tinnitus Handicap Inventory found that tinnitus can cause limitations in tasks of daily living [18]. However, these approaches differ from ours in that they are tested in a clinical setting rather than capturing the patient’s status in their familiar environment. High variations of tinnitus prevalence between countries have been identified in recent works [19]. The aforementioned works can be combined to address the question of whether differences between countries may play a role similar to gender in tinnitus. As we showed in the last paragraph, seasonality is an issue in the tinnitus context,; thus, a combination of country and season seems to be worth further investigations. To raise this research question using mHealth data completes the picture; similar research has not been carried out thus far, and is worth pursuing for the aforementioned reasons. In addition, coming from a broader perspective, the identification of subgroups of tinnitus patients is becoming more and more the subject of investigation [20,21]. Clearly, although selection bias must always be considered in the context of potential subgroups, more and more works have shown that subgroup identification is important for tinnitus research, and that subgroup research must be considered carefully [22]. This work can contribute to subgroup research in general as well as to more specific research question concerning differences between countries and seasons.

For the investigation at hand, the mHealth collection procedure constitutes an issue that has to be considered. The reason is that a large variety of collection methods, strategies, and concepts can be used. We used the strategy called Ecological Momentary Assessments (EMAs) [23], chosen for two reasons: EMAs are carried out in real life (as opposed to a clinical environment) and at arbitrary points in time to capture the moment of a participant. mHealth apps are the major instrument to operationalize EMAs [24,25,26]. Although valuable data sources were established by the latter works, they come with drawbacks [27]. For example, it must be ensured that biases arising through the use of smartphones and the EMA setting are considered. For example, if users have to fill out questionnaires multiple times in a day, they could tend to fill them out only to accomplish the task itself, without providing their actual momentary tinnitus situation. According to our own analyses, when patients consciously deal with their tinnitus on a daily basis, this does not increase their tinnitus perception as a whole [28]. In order to mitigate this, new concepts such as Digital Phenotyping (DP) [29] have been proposed. DP quantifies the human phenotype in a moment-to-moment fashion using active as well as passive data from mobile devices to combine different perspectives. Hence, outliers can be better detected and addressed. As another mitigation approach, the data science community constantly proposes new investigation ideas [30].

For this paper we used the TrackYourTinnitus platform (TYT), which is based on mobile crowdsensing techniques [31], along with EMAs [32]. Crowdsensing connects a group of people who have mobile devices with sensing and computing capabilities collectively sharing data and extracting information in order to measure, map, analyze, and estimate any processes of common interest. TYT was developed to investigate questions about the heterogeneity of symptom profiles of tinnitus patients [26,28,33]. The procedure users of TYT go through is described in [34]. In a nutshell, users register to the platform (website or mobile apps), then have to fill out three baseline questionnaires before they can start the daily EMA procedure. The baseline questionnaires collect demographic information, as well as the anamnestic tinnitus aspects of the users. Importantly, users have to fill out these questionnaires before they are able to start the daily EMA collection procedure. The latter is applied through two native apps, which are available for iOS and Android in the official app stores. The EMA procedure consists of a daily questionnaire with up to eight questions. This questionnaire is applied using two strategies. The first one is based on the idea that users can fill out the questionnaire whenever they want to. The second strategy is based on notifications, which are used to remind users to fill out a questionnaire. Up to twelve random notifications or a fixed scheme can be selected by TYT users. The main scheme used by users is random notification [35]. Note that random notifications mainly follow the idea of in situ measurements, a major goal of EMAs. Of further importance, the setting of TYT has motivated over 8000 users from all parts of the world to provide more than 100,000 questionnaires. The use of mHealth in this context, apart from TYT, has been proposed by many other mHealth projects [6,36,37]. To conclude, the reported facts emphasize that strategies such as EMA or digital phenotyping are promising both in the context of tinnitus research and for other medical entities as well.

The investigation of differences across seasons and country of origin have been identified as feasible using the TYT data source [35]. Therefore, we have decided to raise the following three major research questions (RQ), which to the best of our knowledge have not been considered previously:RQ1: Can momentary tinnitus (Question 1 of the daily EMA questionnaire; yes/no answer options) of TYT users be predicted (binary classifier trained) based on the following features: country, season, age, and sex from the baseline questionnaires as well as the following items of the daily EMA questionnaire: mood, arousal, stress, concentration, and the worst symptom perception. Note that, except for the prediction targets, the features used have been used in other analyses and have proven prediction power on this data set. To be more precise, we excluded features of TYT that are highly correlated with the target, such as tinnitus loudness, tinnitus stress, and momentary tinnitus. However, we included features that are known to be correlated with tinnitus, such as sex and age [16].RQ2: Can the reported tinnitus loudness of TYT users (Question 2 of the daily EMA questionnaire; slider question) be predicted (regressor be trained) based on the same features such as those used in RQ1?RQ3: Based on inferential statistics, are we able to reveal further country- and season-specific differences for the reported momentary tinnitus based on the daily EMA questionnaires of TYT users?

Regarding RQ1 and RQ2, we present results from two machine learning analysis. As TYT was able to gather more than 100,000 EMA questionnaires since 2013, which are comprised of many dimensions, we decided to answer RQ1 and RQ2 based on machine learning algorithms. As we already revealed interesting results on TYT EMA data based on machine learning [38], including the fact that the use of machine learning has been generally recognized in the context of mHealth data in the last years with much attention and valuable results [39,40,41,42], the present paper links these findings.

Regarding RQ3, we present descriptive statistics about the identified country- and season-specific differences. We have detailed the research question into four sub-questions. RQ$3_{1}$ and RQ$3_{2}$ should add more context to the machine-learning results, and are therefore tailored RQ1 and RQ2. RQ$3_{3}$ is a combined perspective of the country and the season, while RQ$3_{4}$ is inspired by medical experts. The following list presents the four sub-questions:RQ$3_{1}$: Are there country-specific differences in momentary tinnitus?RQ$3_{2}$: Are there season-specific differences in momentary tinnitus?RQ$3_{3}$: In light of a combination of country- and season-specific differences, the question arises whether momentary tinnitus varies within the year and across countries.RQ$3_{4}$: Another question is whether country- and season-specific differences in the reported worst symptom can be identified.

Three additional notes are important regarding RQ$3_{1}$–RQ$3_{4}$. First, the last sub-question was set up with the involvement of medical experts, as severe symptoms play an important role in the context of tinnitus research. As TYT asks about nine possible worst symptoms, we investigated how the worst symptom differs across countries and seasons. As the combined perspective taken for RQ$3_{3}$ was useful, this combined perspective was accomplished for RQ$3_{4}$. Second, in the context of season-specific differences, we added an additional dimension, the temperature course throughout the year, inspired by the results of [8].

## 2. Materials and Methods

**The questionnaires.** For the tinnitus prediction task, three linked datasets were used. The first, desginated (1), refers to the baseline questionnaire called the *Tinnitus Sample Case History Questionnaire (TSCHQ)*. The second, designated (2), refers to the daily questionnaire that asks for information about the user’s current sense of well-being. The third dataset, (3), contained information on the temperatures in a country over the annual cycle.

Datasource (1) was the *TSCHQ* questionnaire, which was completed by each TYT user *once* when starting the app for the first time. In this questionnaire, demographic data as well as data about the individual course of the user’s tinnitus are collected, such as the onset of tinnitus and the worst symptom related to tinnitus. There was no purposive selection of individuals, meaning that this was a convenience sample. This implies that we cannot differentiate between users with primary and secondary tinnitus.

When logging in to the TYT platform, users were asked for their worst tinnitus symptom. This symptom can be one of the following:I am feeling depressed because of the tinnitus.I find it harder to relax because of the tinnitus.I have strong worries because of the tinnitus.Because of the tinnitus it is difficult to follow a conversation, a piece of music or a film.Because of the tinnitus it is hard for me to get to sleep.Because of the tinnitus it is difficult to concentrate.Because of the tinnitus I am more irritable with my family, friends and colleagues.Because of the tinnitus I am more sensitive to environmental noises.I don’t have any of these symptoms.

As we recorded fill-in dates of answers to this questionnaire and the country of the user, we were able to link the worst symptom to both the season and country. To assign the fill-in date to a season, we used the astronomical seasons as a guide. More specifically, spring starts on 21 March, summer on 21 June, autumn on 23 September, and winter on 21 December. For countries in the southern hemisphere, the seasons are opposite, i.e., spring becomes autumn, summer becomes winter, etc. As 3.2% of the collected data comes from countries in the southern hemisphere, the correction of the seasons only concerns the analysis for the worst symptom. For the machine learning part, (RQ1, RQ2), countries in the southern hemisphere were not involved due to an insufficient number of completed questionnaires.

The second datasource, (2), refers to the *daily questionnaire*. It included eight questions about the current tinnitus state, i.e., the tinnitus situation and the feelings of the individual *right now*. The eighth *dynamic* question depended on the worst symptom of the individual from the TSCHQ questionnaire, and asked whether the individual has this specific worst symptom right now or not. If an individual answered *I don’t have any of these symptoms* in the beginning, no eighth question appeared in the daily questionnaire. Consequently, the amount of data for Question Eight depends on the number of individuals that selected this worst symptom in the TSCHQ questionnaire. On the other hand, the number of answers for Questions One to Seven equaled each other. These questions can be found in Figure 1, except for Question Three, (*How stressful is the tinnitus right now?* This question was been used as a ML features as it is highly correlated with the targets, and therefore would not bring new insights into the research questions.

Depending on the features selected for the classification task, the number of examples *m* depends on the dynamic question eight. The questions for *mood* and *arousal* are questions using a self-assessment scale (SAM) [34], with nine possible values. Depending on a user’s operating system, the answer is stored with different accuracy. Therefore, rounding errors can occur in the hundredths range on Android phones. We neglected these rounding errors in pre-processing considering the amount of 18 other features (country, season, sex, age, mood, arousal, stress, concentration, worst symptom perception). Note that countries and seasons are categorical and thus one-hot encoded features, as described in Figure 1.

The third dataset, (3), contained information about the temperature in different countries by season. The dataset was crawled from Wikipedia and was originally a list of cities with their average monthly temperatures. The respective country is noted for each city, which means that many countries are represented by several cities. In this case, the data were grouped by country and aggregated on the average level. The weather data in the cities themselves were taken from various weather services in the respective countries, which sometimes resulted in temporal differences in the data, which we consider negligible, due to gradual climate change. The temperature dataset can be found on GitHub (https://github.com/joa24jm/tinnitus-country, accessed on 7 July 2022). To the best of our knowledge, the correlation between tinnitus with temperature has not previously been studied in this way.

### 2.1. Data Preprocessing

The data were taken from from the TYT database [35], which included 8685 registered users as of February 2021. The baseline questionnaire was filled out by 3700 users. Finally the daily questionnaire was answered at least once by 3044 users, resulting in a total of 98,074 answersheets. We can see from this that with respect to the registered users about one in three completed the daily questionnaire at least once. After preprocessing, the remaining merged data frame had 97,742 answersheets. This data frame was used for the statistical analyses provided in the results section.

**Machine Learning Preprocessing.** For the machine learning task, further preprocessing was required. Gradient boosting machines can only handle numerical data with no missing values. We therefore dropped rows that contained missing values, which affected about 24% of the data. One step considered the imbalanced distribution of the target variable *tinnitus occurrence*, as about 79% of the assessments state *yes*. Any dummy machine learning classifier would therefore simply always predict *yes*, regardless of the input of features, and would have 79% accuracy on average. Under a naive dummy classifier, we subsumed an algorithm that always predicts the majority class. We used the F1-score to correctly assess our binary classifier. The F1-score is defined as the harmonic mean of the sensitivity (known as the recall) and precision, calculated as follows: $F_{1}=2TP/\left(2TP+FP+FN\right)$, where TP = True Positives, FP = False Positives, and FN = False Negatives. While the performance can be measured better using the F1-score, the classifier would be overfitted on positive examples. We therefore bootstrapped negative examples with replacement until we had a balanced dataset. We call this the **upsampling approach**. Therefore, the final dataset had 118,054 samples for the machine-learning tasks.

**Estimation of feature importances.** The values in Figure 2 were calculated using three different methods, namely, Gini importance, permutation importance, and correlation metrics. Depending on the feature scaling, two different correlation metrics were applied. If the input feature was categorical, Corrected Cramér’s V [43] was applied. If it was continuous, the Point Biserial method [44] was used. Cramér’s V can have values from 0 to 1, with 0 = no association and 1 = complete association, whereas the Point Biserial correlation can have values from −1 to 1, with −1 = strong negative association, 0 = no association, and 1 = complete positive association. Nevertheless, in order to be able to order the results *within* the column, we took the absolute value from the Point Biserial result. Although all results are in percentages, it is not possible to compare them line by line due to the different units of measurement. Therefore, we created a ranking; for the Gini and permutation importances, both methods are used with the trained gradient boosting machine. The Gini importance is an impurity-based method. The higher it is, the more important the feature is. Notably, within this column all of the values add up to 100%. The importance of a feature is calculated as the reduction of the impurity caused by this feature. For the permutation importance, the percentage values are an estimate of the increase of the error rate on average if that feature were replaced by a random feature. This means that if the variable gender would be replaced with a random variable, the error would increase by 6.43%-points. The column does not necessarily add up to 100%.

### 2.2. Gradient Boosting Machines for Classification and Regression

We choose the Gradient Boosting Machine [45] because it is a tree-based machine learning algorithm related to Random Forests. Machine learning contests on the Kaggle platform have recently shown that this algorithm is superior to most state-of-the-art Deep Learning methods when it comes to tabular data, such as house pricing prediction problems. Both Random Forests and Gradient Boosting Machines use several trees to predict an outcome; however, one of the main differences between these two algorithms is the *time aspect*. This means that the Gradient Boosting algorithm learns from previous misclassified samples by weighting them more heavily. Furthermore, it does not easily tend towards overfitting, as decision trees do.

The whole dataset was divided into three sets, namely, training, development, and testing. Training plus development received 70% of the whole dataset, while testing received the remaining 30%. In all, 1280 combinations of the hyperparameters were evaluated systematically; the final chosen setup can be found provided on GitHub (https://github.com/joa24jm/tinnitus-country, accessed on 7 July 2022). Each combination was cross-validated within the training set using a five-fold split.

### 2.3. Features for RQ1 and RQ2

We used four different groups of features. The first group of features were dummy features indicating whether an individual was from that country or not. As 111 countries would lead to an unnecessary increase in the size of the features, we only used those ten countries with the most filled-out daily questionnaires. These countries were DE, US, NL, CH, GB, CA, RU, AT, IT, and NO. The second group of features were the four seasons. The third group contained age and sex. Note that we did not include the questions tinnitus loudness, momentary tinnitus, and tinnitus stress level as features, as they are highly correlated with the respective target and would not reveal new insights. The last group of features contained information about momentary mood, arousal, stress level, and concentration. This resulted in a data frame with twenty features, one binary target, and 74,360 samples from 2179 users.

## 3. Results

In this section, we present the results for the research questions. At first, we focus on the classification of the first question of the daily TYT questionnaire, (*Did you perceive the tinnitus right now?*). We refer to this question as momentary tinnitus in the following. Second, we consider the regression of the tinnitus loudness, (*How loud is your tinnitus right now?*) and refer to this question as tinnitus loudness. Third, we analyze these two targets, momentary tinnitus and tinnitus loudness, in a global context by relating them to the country, season, and temperature.

### 3.1. RQ1: Is the Momentary Tinnitus of TYT Users Predictable Using the Features Country, Season, Age, Sex, and from the Daily EMA Questionnaire, Mood, Arousal, Stress, Concentration, and Worst Symptom Perception?

The Gradient Boosting Machine attained a final **F1-score of 93.79% in the testing set**. The confusion matrix reveals *True Negatives = 17,351*, *True Positives = 15,953*, *False Negatives = 358*, and *False Positives = 1755*. When leaving out the features sex and age, the mean test score dropped to 88.9% using the same hyperparameters. Using only the binary features seasons and countries lead to a strong decrease of the score on the test set down to 58%. This is caused by the low dimensional feature space.

### 3.2. RQ2: Is the Reported Loudness of TYT Users Predictable Using the Same Features as in RQ1?

Regarding second research question, we wanted to estimate the tinnitus loudness based on the features listed in Figure 1. We tried to optimize the Gradient Boosting Regressor for absolute deviation from the estimated loudness to the true loudness. We refer to this measure as abs_mean_error. In contrast to momentary tinnitus, there was no skewed distribution with respect to tinnitus loudness. This did not, in our estimation, produce a need to generate samples to produce, for example, a Gaussian distribution of the true values. We chose to train the regressor on the mean absolute percent error rate because this measure directly provides a sense of how well or poorly the regressor is performing. In each case, the regressor underestimates the marginal regions (<0.2 and >0.7) of the reported loudness and slightly overestimates the middle regions. On average, it is off by eight percentage points. Thus, if a user reports a loudness of 70%, the regressor estimates a loudness between 62 and 78% on average. A density distribution of the reported loudness and the estimated loudness is shown in Figure 3. Users in the marginal areas tend to be underestimated by the regressor (loudness from 0.0 to 0.2 and 0.7 to 1.0). In the middle ranges (loudness from 0.2 to 0.7), they tend to be overestimated. Nevertheless, a mean absolute error of 8.1% was achieved on the development set (std = 0.0006) and an error of **7.9%** on the test set.

#### Feature Importance

In order to find out which of the variables had a high impact on tinnitus prediction, we looked at the feature importance of the Gradient Boosting Machine. In order to determine the feature importance more accurately, we investigated three methods. The first is called the *Gini importance*, the second the *permutation importance*, and the last the *correlation*. These three methods measure the feature importance in different units, which makes it impossible to compare importance between methods. However, in order to make the results comparable we created an importance ranking. The lower (i.e., greener) the ranking number is, the more important the feature is for the model to predict the target. Figure 2 shows the features with their importances for the models. We calculated the feature importance for each model (classifier and regressor) separately. In order to classify the momentary tinnitus, demographic features are most important, with an average rank of 4.5. To regress the tinnitus loudness, the daily questions are most important with an average rank of 4.16). For both models, age is the most important feature (average rank = 2), as it has a high cardinality. Conversely, country has a lower importance (average rank = 13.6), as it has only a low cardinality with low variance. For each model, the Gini importance values within one column add up to 100%, while the permutation importance indicates the absolute increase of the error rate if that feature was left out. As the percentages cannot be compared between columns, only within a column, the ranks of the feature importances are provided. As age is a feature with high cardinality, it clearly helps the tree-based Gradient Boosting Machines to predict the targets. The high feature importance for the age variable could be an indication of overfitting of users with many assessments. The permutation importance of 29.7% suggests that the performance becomes 29.7% percentage points worse when age is replaced by a random variable. For example, almost all Russian users consistently answered the question about current tinnitus in the affirmative. Within the countries feature, Russia therefore has a high correlation with current tinnitus. However, because there are relatively few users compared to all users, the Gini importance for Russia only shows a value of 2.19%.

### 3.3. RQ3: Are We Able to Reveal Country- and Season-Specific Differences for the Reported Momentary Tinnitus Based on the Daily Questionnaire of TYT Users?

To answer this question, we used 97,742 responses from 3691 users from a total of 111 countries for the period April 2014 to February 2021. For further analysis, we restricted ourselves to the countries represented by at least 30 users and with more than 300 questionnaires in total. For this subset of 15 countries there were 3163 remaining users with a total of 88,049 filled out daily questionnaires. The most responses were from Germany, with 51,804 completed questionnaires (58.84%) generated by 1410 users (38.20%), whereas the fewest completed questionnaires came from the Federative Republic of Brazil, with 334 completed questionnaires (0.38%) generated by 50 users (1.35%). The mean number of filled-out questionnaires per country was 5870 (std = 13,058). The mean number of users was 210 (std = 357). For the question of interest, *Did you perceive the tinnitus right now?* (question1), the mean value for ’Yes’ was 78.97% (std = 12.21%), an interquartile range of 15.73%, with a maximum value of 95.58% from Italy and a minimum value of 48.66% from Norway.

RQ$3_{1}$: Are there country-specific differences for the momentary tinnitus?

A chi-square test of independence showed significant differences between the countries, $\chi^{2}\left(14,n=85933\right)=2441.44,p<0.001$ in momentary tinnitus; 105 post hoc $\chi^{2}$ tests were performed to compare pairwise differences. Using corrected *p*-values, 91 pairs of countries were rejected (*p* = 0.05). Fourteen pairs could not be rejected at *p* = 0.05, i.e., the pairs Germany–Great Britain and Germany–Sweden. This indicates that these countries have a similar pattern in momentary tinnitus occurrence. A detailed overview of the answers to question 1, (*Did you perceive the tinnitus right now?*), is shown in Figure 4.

For the countries CH, DE, GB, NL, and US, we looked at momentary tinnitus broken down by gender. For these countries, women report a probability of 69% of having momentary tinnitus, whereas men report a probability of 78% of having momentary tinnitus. For the Netherlands, it is striking that for females the Yes–No ratio is almost equally distributed, with 53% to 47%. For male users from the USA the probability of currently having tinnitus is the highest, at 88.5%.

In order to ensure comparability between the countries under consideration, we examine the demographic variables in detail in Figure 5. Additionally, the data are grouped by gender. The $\chi^{2}$ for *handedness* and *family history* is n.s. For the age distributions, the post hoc Tukey test shows significant mean differences for DE and the US (*p*< 0.05) and for GB and the US (*p*< 0.01). The figure supports the comparability of the five countries that are mainly discussed in RQ3.

RQ$3_{2}$: Are there season-specific differences for momentary tinnitus?

To answer this question, we again analyzed only countries represented by more than thirty users with more than 300 completed questionnaires *per season*. This filter setting holds true for Switzerland, Germany, the United States, Great Britain, and the Netherlands. The largest sample is again for Germany, with 51,534 completed questionnaires, while the smallest sample for the UK, with 3684 completed questionnaires.

If we do not group by country, it can be seen that the greatest probability for momentary tinnitus is in summer, with 83.4% (std = 8.6%). In contrast, the lowest probability for momentary tinnitus is in winter, with 71.0% (11.8%). The interquartile range is 14.5% for winter and 11.8% for summer. If we group by country, the highest probability for momentary tinnitus is in summer in Great Britain (95.7%), while the lowest in winter in Switzerland (60.7%). The ratios of yes–no responses can be found on GitHub (https://github.com/joa24jm/tinnitus-country, accessed on 7 July 2022). Considering both these five countries and 111 countries in the present data set without setting a questionnaire or user threshold, the probability of momentary tinnitus perception is 80.6% in summer, 80.1% in fall, 78.6% in spring, and 75.1% in winter. A $\chi^{2}$ test of independence showed a significant association between season and momentary tinnitus for all countries without a user or questionnaire threshold, $\chi^{2}\left(3,n=95,446\right)=216.19,p<0.001$. Overall user reporting for tinnitus is thus most likely in summer.

Baseline characteristics from this questionnaire for the five countries (CH, DE, GB, NL, US) as well as all other countries can be seen in Figure 5. These five countries are the subject of our RQ$3_{2}$. To ensure comparability between countries, we considered other demographic data in more detail. For the characteristics *handedness* and *family history of tinnitus complaints*, a $\chi^{2}$ test was performed. The $\chi^{2}$ test showed that there was no significant association within the country groups, $\chi^{2}$(8, n = 2319) = 6.64, *p* = 0.58 for *handedness*, and $\chi^{2}$(4, *n* = 2314) = 4.33, *p* = 0.36, for *family history*. To compare the age distributions between countries, a one-way ANOVA was performed with F(4, 2267) = 5.17, *p*< 0.001. A post hoc pairwise Tukey test revealed differences between DE and US (mean diff. = 2.36, *p*< 0.05) and between GB and US (mean diff. = 5.07, *p*< 0.01). The remaining eight pairwise groups had no significant differences in their means.

In a slightly different approach, we considered months instead of seasons. In addition, we examined the respective average temperature per month in relation to tinnitus occurrence for the countries considered (i.e., Switzerland, Germany, U.S., Great Britain, and the Netherlands). In this context, a positive correlation means that the higher the temperature, the more likely there is to be momentary tinnitus. A high positive correlation was obtained for the Netherlands (r(10) = 0.83, *p* < 0.001), Great Britain (r(10 = 0.86, *p* < 0.001), and Switzerland (r(10) = 0.72, *p* = 0.009). For Germany and the U.S., however, the correlation between temperature and tinnitus occurrence can be considered uncorrelated (both *p*-values > 0.1). The cyclical temperature pattern associated with tinnitus over the year for the various countries is shown in Figure 6. The larger the circle is, the higher the average probability of momentary tinnitus for this country in this month is. The size and color of the cycles indicate the chance of momentary tinnitus. There was a statistically significant difference between the countries as determined by one-way ANOVA (F(4, 55)= 6.69, *p*< 0.001). A post hoc Tukey’s test indicated that the annual course of momentary tinnitus is different between the country pairs Netherlands–U.S. (*p*< 0.01) and Switzerland–U.S. (*p*< 0.01).

RQ$3_{3}$: In the Light of a Combination of Country- and Season-Specific Differences, does Momentary Tinnitus Vary within the Year and Across Countries?

In contrast to the previous section, we ignore temperature in this question. Instead, we examine the following: For each of the countries considered, and for each individual month of the year, we calculated the probability of tinnitus by dividing the number of yes responses by the sum of responses. In the following step, we examine the probability of tinnitus over the course of the year. To increase comparability, we additionally calculate the average of the tinnitus probability for all available data on a monthly basis.

As most of the data come from Germany, this country has a correspondingly large influence on the average values. Accordingly, the curve for Germany is very similar to the curve of all data (statistic = 0.17, *p* = 1.00). On the contrary, the Netherlands, the U.S., and Switzerland reveal a different distribution of tinnitus, with *p*-values < 0.01. For Great Britain, the distribution can be considered to be slightly different, as the *p*-value is 0.10. An overview of the distributions compared with the average is shown in Figure 7. The graph indicates that people in different nations perceive tinnitus differently throughout the year. A summarizing statistical overview is provided on Github (https://github.com/joa24jm/tinnitus-country, accessed on 7 July 2022).

The highest probability of tinnitus is in the US, with an average chance of 87%, while the lowest probability is in Switzerland, with 68%. The largest variance occurs in Great Britain, with 16% standard deviation, and the smallest in Germany, with 4%. For this data set, tinnitus occurred the least in Switzerland in March (53%), and most in the UK in August (98%).

RQ$3_{4}$: Can Country- and Season-Specific Differences in the Reported Worst Symptom be Identified?

In order to answer this research question, we again focused on the five countries CH, DE, GB, NL, and US. When registering on the TYT platform, the question about the worst tinnitus symptom is asked once. For each country and season, we calculated the relative number of answers within a country to compare which symptom is more likely in which season. Each column adds up to 100%. The 1310 users from Germany had the lowest standard deviation (0.94 std). The Netherlands, with 175 users, had the largest standard deviation (2.01 std). *I find it harder to relax* was the most likely symptom in the Netherlands in fall, with 8.57%, and, at the same time, with a global maximum. *Feeling depressed* ranked second for the UK and the Netherlands. For the U.S., the two worst symptoms were *difficulty following a movie or conversation* and *concentration problems*. For the U.S., however, there was little variation between seasons within these two worst symptoms. *None of these symptoms* ranked second for Switzerland. *Irritability with friends and family* was the least indicated worst symptom for all countries. A chi-square test was performed between distribution of the worst symptom and country. There was no statistically significant relationship between worst symptom and country, $\chi^{2}$(40, *n* = 6) = 0.53, *p* = 1.0. Details are shown in Figure 8.

In a similar approach, we disregarded countries and investigated the evolution of the worst tinnitus symptom between seasons. Thus, we examined whether there were different worst symptoms per season. *Because of the tinnitus I am more irritable with my family, friends and colleagues*, was the most unlikely symptom (mean = 5.9%, std = 1.0%. The most likely symptom was *I find it harder to relax because of the tinnitus* (mean = 17.7%, std = 1.9%). Details are provided on GitHub (https://github.com/joa24jm/tinnitus-country, accessed on 7 July 2022). Difficulties in relaxing was the worst symptom across all seasons. The data further indicate that feelings of depression were stronger in the months of autumn and winter. Difficulties in following conversations were more pronounced in summer. Irritability with colleagues or family was the least selected symptom. However, a chi-square test of independence showed that there was no significant association between worst symptom and season, $\chi^{2}$(24, *n* = 3458) = 30.86, *p* = 0.16.

## 4. Discussion

Research on the temporal fluctuations of individually perceived tinnitus is scarce and the underlying mechanisms of these fluctuations are poorly understood. Here, we present an analysis of a large longitudinal dataset that highlights the importance of home country and the season of the year for understanding dynamic changes in tinnitus. To estimate the magnitude of the country- and season-specific influence on momentary tinnitus, we conducted a feature importance analysis (Figure 1), which revealed that age has the strongest influence on tinnitus fluctuations, followed by stress, concentration, and emotions. Seasonal influences are much weaker compared to these influencing factors. The influence of home country shows the weakest influence. Nevertheless, home country and season seem to have an influence on the tinnitus perception of affected individuals. A further analysis including the average temperature of the respective countries over the year revealed meaningful correlations between momentary tinnitus and the temperature for the Netherlands, Great Britain, and Switzerland to suggest that there might be a biological mechanism that moderates or mediates tinnitus perception based on the temperature. Further research is needed to investigate this research question in more depth.

The observed results of the presented analysis are, in detail:For RQ1, we found that we can predict momentary tinnitus with an F1-score of 93.79% on the assessment level.For RQ2, we found that the tinnitus loudness can be regressed with a mean absolute error rate of 7.9%-points on a scale from 0 to 100%.For RQ$3_{1}$ (country specific differences for the momentary tinnitus), we found that most countries report momentary tinnitus differently.For RQ$3_{2}$, we found season-specific differences in momentary tinnitus. If the data are not grouped by country, momentary tinnitus is most likely to occur in the summer. This is in contrast to the results of [7], where Tinnitus was most likely in summer. When we group our data by country, an ambiguous picture emerges between countries as to the most likely season for tinnitus.For RQ$3_{3}$, we found that the momentary tinnitus does vary within the year and within countries. We found that this momentary tinnitus variance within one country is different from one country to another, i.e., if we compare Great Britain to the US.For RQ$3_{4}$, we examined whether the distribution of the worst symptom changes between years or whether it is significantly different between countries. Our analysis showed that neither varied significantly, although the numbers suggest small differences.

We first summarize the importance of the results, then discuss several of them in more detail.

In our TYT data set, the country of users or the season in which an assessment was provided revealed a small prediction power for momentary tinnitus. The prediction power is better than guessing, and the statistical analyses showed differences between countries and seasons as well as within countries. This indicates that country- and season-specific differences should be considered when explaining inter- and intra-individual variance in tinnitus fluctuations.Self-reported mHealth data, which are collected globally with a longitudinal design, can contribute to the understanding of the biological mechanisms underlying tinnitus.

Although we found significant differences for momentary tinnitus between seasons and countries, this does not establish causality between the features and the target. There are a few limitations that need to be discussed. First, there might be a myriad of other reasons why tinnitus is more likely in certain countries in summer and in others in winter. Influencing factors could be, for example, air pressure, stress level, or the number of hours of sunshine. It could be that unknown information, such as hearing aids or cochlear implants, distort the loudness of the tinnitus. In addition, a certain population might be predisposed to hearing loss due to individual work or living conditions. Second, user numbers varied widely between countries. This can lead to a selection bias in the evaluation. Consider the filter criterion, here, “at least 30 users per country”. If one user was particularly active in filling out the daily questionnaire and the other 29+x users were not, this might lead to a selection bias. Third, although our research results indicate different seasonal trends in momentary tinnitus in different countries, there may be individuals who perceive tinnitus seasonally quite differently, possibly even completely in the opposite direction. This means that these findings are not applicable to these individuals. Fourth, data about comorbidities (e.g., diabetes, Ménière’s Disease, hearing loss) where not collected in this study. If a participant suffers from tinnitus plus one or more other diseases, seasonal fluctuations in the other disease might influence the perception of tinnitus. Fifth, the use of corrective devices for hearing loss, such as hearing aids, cochlear implants or hearables, was not recorded in this study. It is possible that the availability and usage of these devices differs between countries, which might have influenced individual tinnitus perception. That hearing aids or cochlear implants as well as other influences on and in the ear can have an impact has previously been shown in other studies [46]. Therefore, we will include hearing aids or cochlear implants as factors in future studies.

For the worst tinnitus symptom per country and season, comparability between countries and seasons may be biased by selection due to the low number of users per category. For Switzerland, for example, we would expect 3.17 individuals per symptom per season (i.e., 2.8% per line) if symptoms and seasons were equally distributed. In this respect, it is surprising for Switzerland, for example, that *relaxation* is more difficult in summer (7.89%) than in winter (3.51%). The situation is different for Germany. Here, we have a large number of users at 1,310, and would expect 36.4 individuals per category if the symptoms were equally distributed among all seasons. This argument is supported by the fact that the variance in Germany is lower than in Switzerland. Nevertheless, we can observe for Germany that *relaxation* is more difficult in spring and autumn (about 5%) than in summer or winter (about 3%).

### 4.1. Performance Depends on Which Level We Split: Assessment Level vs. User Level

By stratifying at the **assessment level** (i.e., on the level of filled-out questionnaires), one can ensure that the distribution of the target between test and training data remains the same. The specific problem at hand is that several users have filled out different numbers of assessments. There are many users with only one or two assessments, and a few users with several hundred or thousand assessments (so-called power users). These power users are highly likely to be present in the training, development, and testing data. Any model is therefore predestined to overfitting on these power users. This can be addressed by excluding users that are in the training set from the test set and vice versa. We then no longer evaluate at the assessment level, but at the **user level**. However, the F1-score in the test set then drops from 93.79% to around 50–60%, depending on different test sets with different power users.

However, if a split is performed on the user level, then other implications arise. For example, there are features that are user-dependent, and which therefore reduce the number of learnable parameters in the model when splitting the data at the user level. In our case, for example, country, gender, age, and season are such features. To be more specific, if there are, for instance, only German users in the training data, and only English users in the test data, then the feature country has no more variance and therefore no prediction power for the model. As another example, if a male user who is 43 years old reports momentary tinnitus as “Yes” several hundred times, then the model learns that 43 year old males always have tinnitus. However, this would have nothing to do with the dynamic assessments, and therefore contradicts the idea of Ecological Momentary Assessments. This would partially explain the drop in F1-score between the training and test sets. We therefore took a subset of the features that we knew retain their variance even when being split at the user level. These features were mood, arousal, stress and concentration. If we split at the assessment level, i.e., allowing the same users to be in both the training and test data, we have an F1-score of  84% in the test set, which is significantly better than guessing. If we now additionally split on the user level, the F1-score drops again to 50–60%, which suggests overfitting of the training users.

The bias in the selection of users remains. A user who has completed many assessments is represented in both the train and test data, which raises doubts about the generalizability of the model, as we may have trained a user-specific model. On the other hand, if we try to stratify for users, the distribution of the target and demographic data are not retained and we would have to collect a much larger amount of data, which is not common in many mHealth studies. In addition, in our opinion, we need more reference points to compare machine learning-based results using mHealth data sources such as this to work on the issue of whether the user level or assessment level is more appropriate. Finally, any stratification technique eventually creates a bias. We decided on the user bias in order to be able to stratify correctly for the target. This allowed us to use more data to train our models. The generalizability of the model to users from a different population is not finally known; however, it is known that the model can make predictions at the assessment level for users who come from a known population. This is shown by the high F1-score of the test set at the assessment level. In the current investigations, we evaluate these differences in more depth.

### 4.2. Feature Importance

High cardinality features such as age and the daily questions are assigned a higher importance, as these features can be easily split into multiple and potentially pure subsets. For binary features, the tree classifier can only split the data once. However, for features with high cardinality, the tree can potentially split the data n_unique - 1 times. Feature importance does not establish causality between input variables and the target; rather, it is an estimator of which variable has the greatest predictive power for the Gradient Boosting Machine. Any other classifier, such as a neural network, would potentially produce a different ranking for feature importance. Among the percentages, the 93.3% permutation importance for age in the regressor model is prominent. The value of 93.3% indicates that the model loses almost all of its predictive power without the age feature. However, as the model was trained and evaluated with the *mean absolute error*, this percentage value cannot be easily transferred to the mean absolute error, and is only an indicator for the importance for the model.

### 4.3. The Temperature Dataset

Although the more than 300 different sources of the individual figures are very well referenced within Wikipedia, there could be noise in the data. For example, only a few cities in a country are a limited representation of the temperature across the country. Second, noise may occur because the temperatures come from different years, some of which were recorded before the EMA data were collected. Nevertheless, we believe in the reliability of the temperature data because temperatures hardly change significantly within a decade in a country. If GPS data are stored by users when filling out EMA questionnaires, it will be possible in the future to determine and further refine weather data more locally. This means that wind speed, air pressure and humidity could be included as additional features.

### 4.4. Worst Season for Tinnitus

We define one season as worse than another if the probability of momentary tinnitus is higher on average. This question cannot be answered unambiguously and conclusively. Related work on tinnitus and seasonality does suggest winter as the worst season [7,9,10]. However, 41.8% of all individuals (*n* = 100) report perceiving summer as the second-worst season, which argues against the theory of seasonal affective disorders [47]. In a study which aggregated tinnitus search requests from online platforms by season and country, winter was highlighted in terms of request frequency [7]. However, the results are different even for countries with similar longitudes. For example, this is the case for Sweden and the United Kingdom. The noise in the results could be due to confounders or to the aforementioned selection bias.

### 4.5. Outlook

In our future work, we are interested in three research directions. First, we plan to compare the results of TYT to other data sources that have similar characteristics. Second, a more in-depth inspection of the user and assessment perspectives of TYT users will be addressed. Third, for country- and season-specific differences, we intend to consider other prediction features that might be of interest. For example, depression [48] can vary significantly throughout the year and between countries; as the hours of sunshine are different in many countries, this type of feature should be investigated.

## Figures and Tables

**Figure 1 jcm-11-04270-f001:**
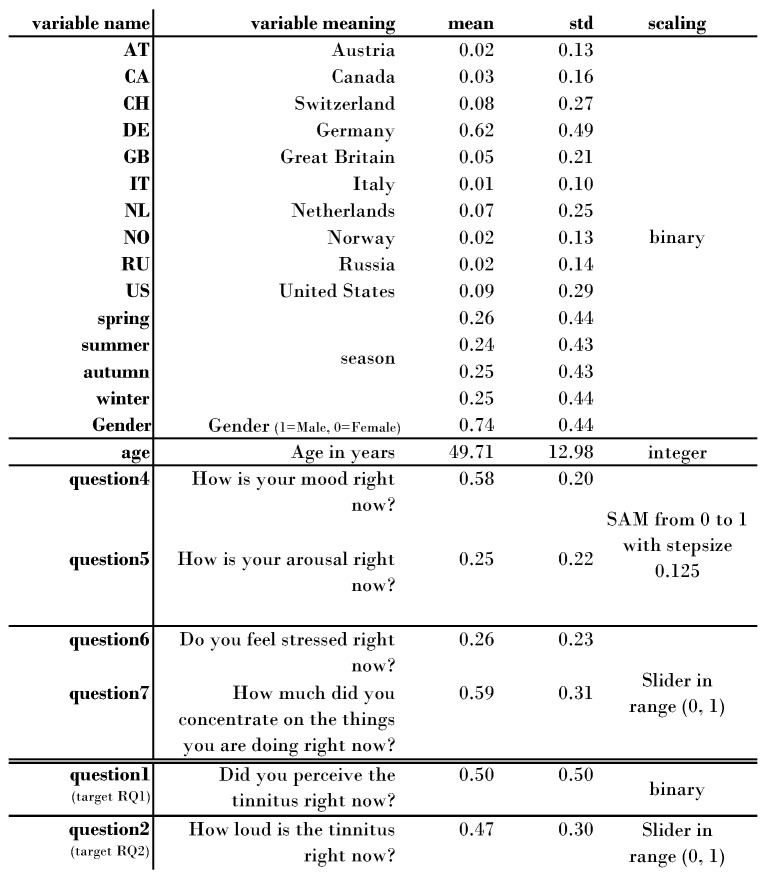
Overview of the features and targets used to train the gradient boosting machines for RQ1 and RQ2. SAM = Self Assessment Scale.

**Figure 2 jcm-11-04270-f002:**
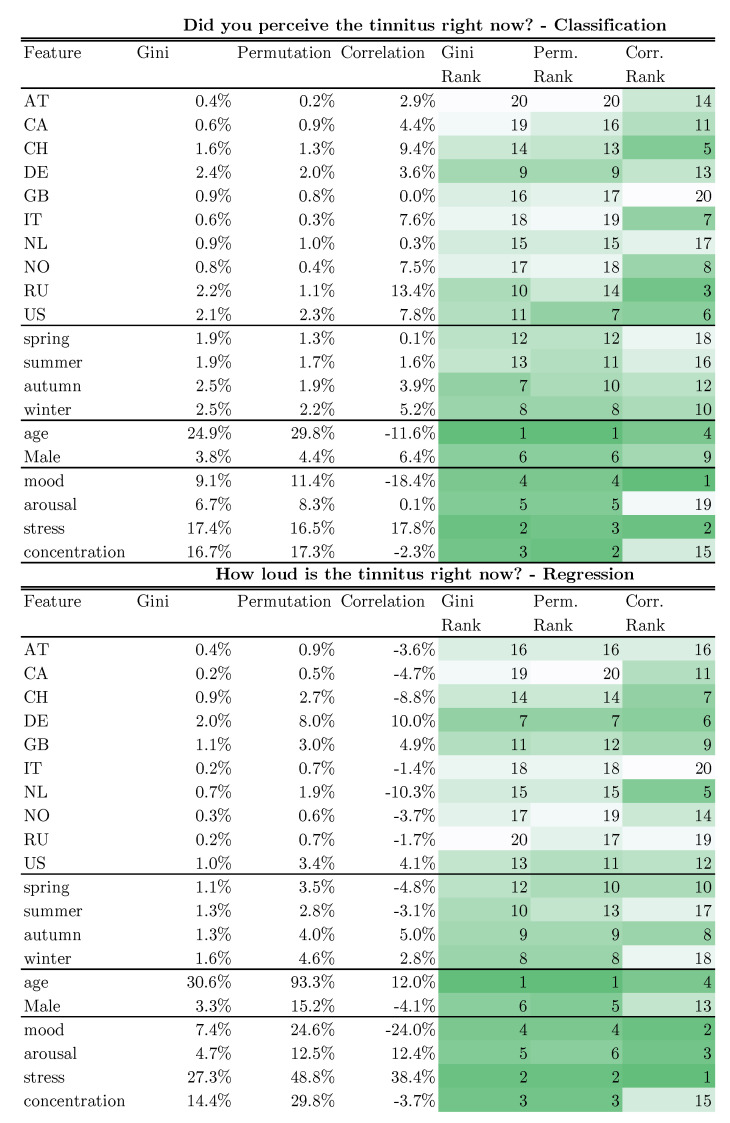
Feature importances of the Gradient Boosting Machines (both classifier and regressor) of univariate features with the two targets momentary tinnitus and tinnitus loudness. Perm = Permuation, Corr = Correlation, AT, CA, CH, DE, GB, IT, NL, NO, RU, and US are ISO2 country codes. The feature is more important the greener the cell is.

**Figure 3 jcm-11-04270-f003:**
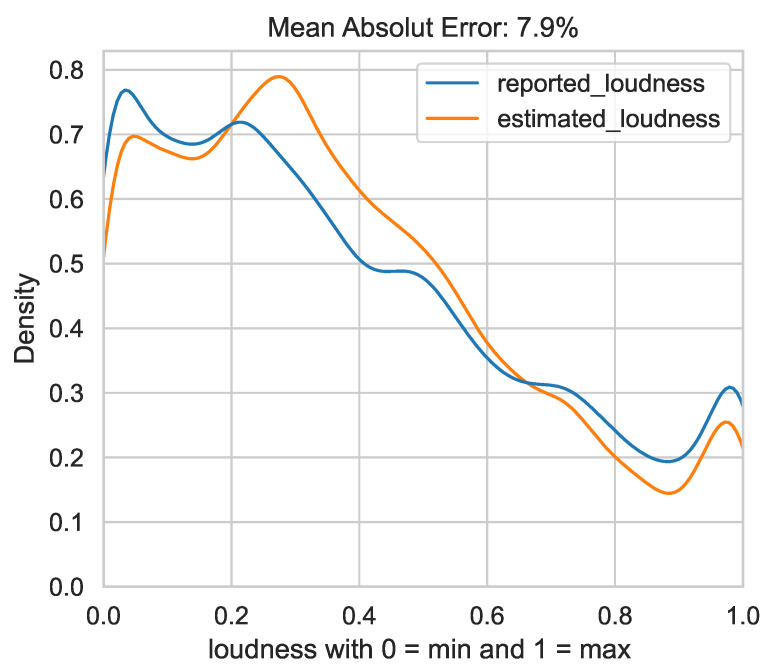
Density curves for reported loudness and estimated loudness for all assessments of the test set.

**Figure 4 jcm-11-04270-f004:**
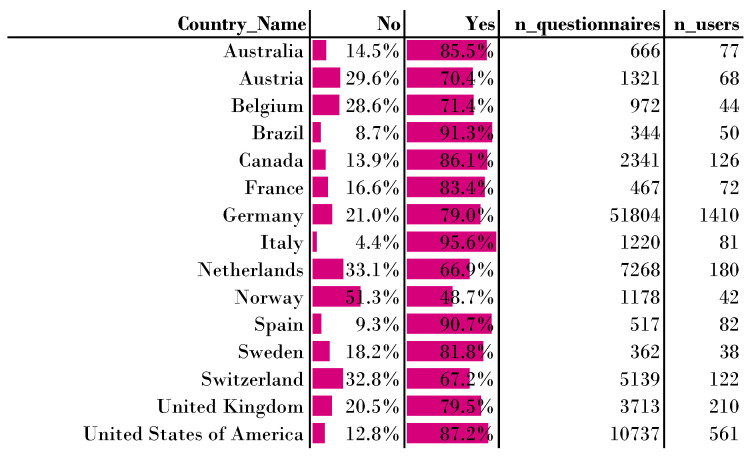
Momentary tinnitus per country for TYT users. The chance of tinnitus occurrence is 78% (std. 12). The longer the bar, the higher the value.

**Figure 5 jcm-11-04270-f005:**
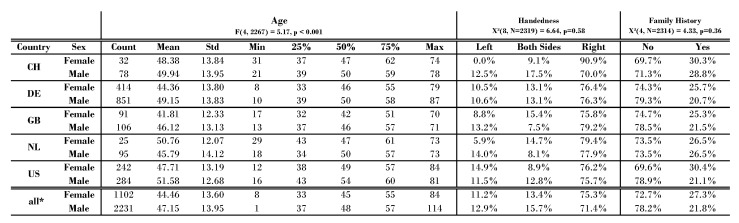
Statistical comparison of five countries, CH, DE, GB, NL, and US, with all users. * These five countries, CH, DE, GB, NL, and US, are included in all countries.

**Figure 6 jcm-11-04270-f006:**
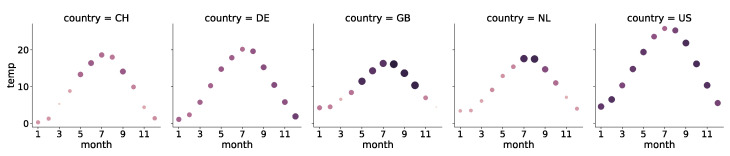
Cyclical temperature pattern for Switzerland (CH), Germany (DE), the United States of America (US), the United Kingdom of Great Britain and Northern Ireland (GB), and the Netherlands. The bigger and darker the cycle, the higher the chance of momentary tinnitus.

**Figure 7 jcm-11-04270-f007:**
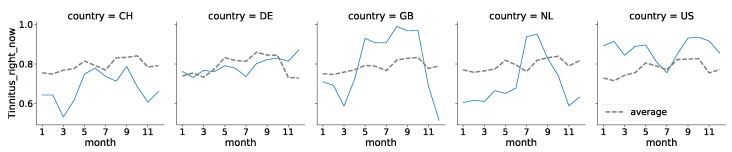
Course of occurrence of tinnitus over the year for Switzerland (CH), Germany (DE), the United States of America (US), the United Kingdom of Great Britain and Northern Ireland (GB), and the Netherlands. The dashed grey lines show the average of tinnitus occurrence for all data *except* the country plotted on this axis, while the blue line corresponds to the country plotted.

**Figure 8 jcm-11-04270-f008:**
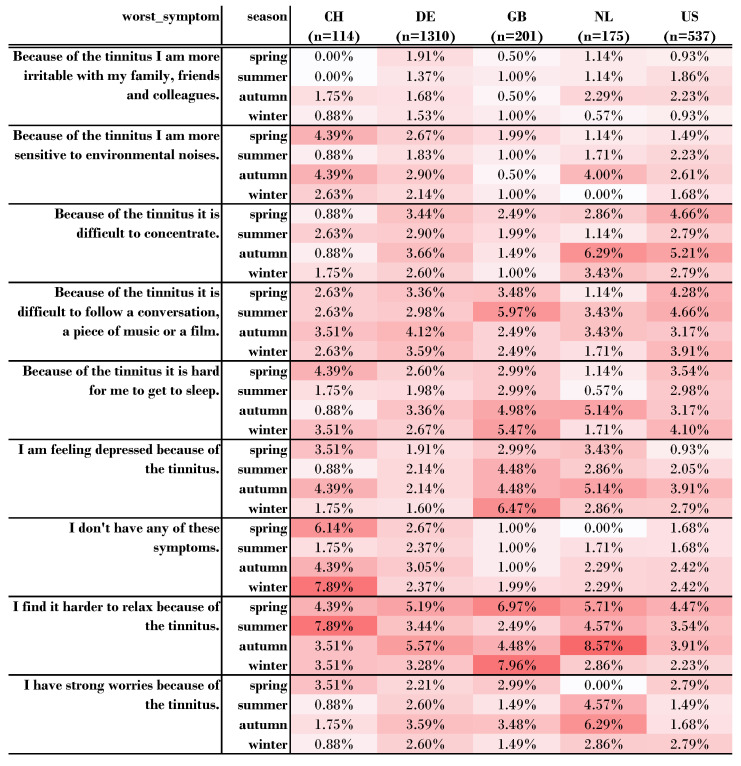
Distribution of the worst symptom for each country and season. We only considered countries with more than 300 questionnaires from more than 30 users. Each column adds up to 100%. *n* denotes the number of users from this country. The redder the cell, the larger the number.

## Data Availability

According to the GDPR, the data to replicate these results are available upon reasonable request to the corresponding author. All code to replicate the results, models, numbers, and figures is publicly available on GitHub (https://github.com/joa24jm/tinnitus-country, accessed on 7 July 2022).

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
