# Peer review of "Prediction of Tinnitus Perception Based on Daily Life MHealth Data Using Country Origin and Season"

_jcm, 2022, doi:10.3390/jcm11154270_

Round 1

Reviewer 1 Report

In the current study the authors have used tinnitus symptom data from TrackYourTinnitus (TYT) mHealth platform and analyzed by machine based learning to investigate the effect of variables including geolocation (country) and weather on momentary tinnitus.  The study concluded that both country origin and weather can influence assessment of tinnitus. Overall this is a fairly well-designed study. However, the authors would require to address the following.

1.       How was the population enrolled in the study? If some participants had history of other diseases such as diabetes or Ménière's disease, how would that influence the overall results?  Tinnitus loudness can also be affected by use of hearing aids or cochlear implants or if a certain population in a participating country had pre-disposition to hearing loss.  How was the effect of those variables accounted for extrapolating the results?

2.       The authors here did account for the weather difference, more specifically effect of temperature difference on momentary tinnitus. They also acknowledge the fact that in a given country there can be regional difference in weather.  Similarly, in a given season, there can be regional differences in wind speed, humidity and atmospheric pressure in a given country and these parameters are shown to impact tinnitus (Schmidt et al., 2017). If the data is combined from various regions, will that affect the final outcome?

3.       The data used for this analysis was collected until February 2021. COVID-19 pandemic has shown to affect tinnitus symptom with various degree. Can including data from COVID pandemic time influence the outcome of this longitudinal  research ?  

Author Response

Please find the point by point response attached.

Reviewer 2 Report

First of all, I would like to thank for reading this manuscript. The information presented in the manuscript is interesting and adds to knowledge; however, there are some issues that should be corrected. See my specific comments below.

Introduction

Generally, in my view, since the main topic of this article is tinnitus, more information should be provided regarding this symptom. E.g., in the introduction, primary/secondary or subjective/objective types are not mentioned. Therefore, I do not find this is an appropriate introduction to this article. Please correct. Moreover, in the results section the parameters regarding momentary tinnitus are analysed, but it is not mentioned what this means (nor in the introduction or methods).

Regarding the influence of tinnitus, only quality of life is mentioned; however, it has a significant impact on everyday functioning. This is particularly important as the questionnaires generally applied in everyday clinical practice (e.g., Tinnitus Handicap Inventory) can detect the consequences regarding everyday tasks. Please include and cite.

[Mancini PC, Tyler RS, Smith S, Ji H, Perreau A, Mohr AM. Tinnitus: How Partners Can Help? Am J Audiol. 2019 Mar 15;28(1):85-94. doi: 10.1044/2018_AJA-18-0046.]

[Mavrogeni P, Maihoub S, Tamás L, Molnár A.  Tinnitus characteristics and associated variables on Tinnitus Handicap Inventory among a Hungarian population. J Otol. 2022, https://doi.org/10.1016/j.joto.2022.04.003.]

Line 50. So far, differences on gender have been found to also play a role for tinnitus research. What kind of influence was detected in that articles? Please specify. Additionally, previous studies have also focused on the possible effect of age on tinnitus, which should be mentioned as well.

Line 73-75. This is also an issue, I agree. Additionally, regarding tinnitus, especially of the chronic types, it is always a problem that ‘monitoring’ the symptoms can worsen the actual symptoms reported by the patients. This is generally a limitation of self-reported questionnaires, especially when patients have to fill in the questionnaires multiple times a day. I think this should be included as a potential limitation.

Material and Methods:

Line 191. These questions can be found in Table 1, except for Question Three. Please, specify the reason why this question cannot be found there.

In the Data Processing section, much information regarding data processing can be found that is essential to interpreting the results. However, I miss some important information. See my specific comments below.

Line 226. The calculation of the F1 score is missing. The readers who are unfamiliar with statistics may cannot correctly interpret the results without this. The potential readers of the journal are medical professionals and not statisticians.

Line 234. Please use the term Cramér’s V.

Line 235. Cramer’s V is defined in range (0,1), whereas the Point Biserial correlation is defined in range (-1,1). Cramér’s V is not defined in range but as category (or nominal values); please correct this. In the previous sentence, it was also stated that Cramér’s V is used in the case of categorical input. Additionally, I miss the interpretation of the categories (i.e., 0 means no association and 1 indicates a complete association). This is also missing regarding the Biserial Correlation ranges. Please include.

Results

My general comment regarding the results section is that it is more like a summarisation of the statistical analysis results but does not include any explanation/interpretation of the results. This is particularly important for readers unfamiliar with statistical analysis and from the authors’ point of view as well (i.e. nobody would like to misinterpret the results). Please rewrite the results section, considering this. Sometimes details regarding the statistical methods can be found in the results section; however, the conclusions based on the results are often missing.

Table 2. Please include the explanations of the abbreviations (i.e.  Perm. Rank, Corr. Rank)

Line 312, Feature Importances. Tinnitus onset was considered in this case (if there was enough data available on this question)? This can be an essential influence factor.

Line 346. Most responses are from Germany.. please also include the % values of these results; it is easier to interpret for the reader.

Line 355. .. that there are significant differences.. Significant differences in what? If we continue reading, it can be concluded that momentary tinnitus occurrence was analysed in this case. For better understanding, please include it at the beginning of this section.

Line 369. I do not recommend using the term ‘handedness’ in this case, as it is generally used for right or left-hand dominance. In this case the authors refer to the side of tinnitus symptom (i.e. right, left or bilateral); hence, the term laterality or localisation is more accurate. Please also correct in Table 5. Handedness also has consequences regarding tinnitus based on previous research results, but this was not analysed in the current investigation.

Discussion

Generally, the current study results are not contrasted to previous research outcomes in the discussion. See my specific comments below. Please revise this section as well.

Line 479. This is in contrast to the result of [11]. What was concluded in that study?

Line 583.  The temperature dataset. Are there any previous research data accessible on this question? Currently, only Wikipedia is referenced, and the results are not contrasted to previous research outcomes.

Line 591. Worst season for Tinnitus? What do the authors think, what can be the possible explanation of their results (if there are any). As referenced, previous research has indicated winter as the worst season. How did the authors of the cited articles explain this result?

Author Response

Dear reviewer,

please find the point-by-point response attached.

Kind regards

The authors

Round 2

Reviewer 2 Report

Thank you for the revised version of the manuscript. The authors made efforts to improve the quality of the manuscript; however, some minor changes are still recommended. Find my specific comments below. 

since the main topic of this article is tinnitus, more information should be provided regarding this symptom. E.g., in the introduction, primary/secondary or subjective/objective types are not mentioned. Therefore, I do not find this is an appropriate introduction to this article. Please correct.

Thanks a lot for this idea, which certainly helped to improve the introduction. We defined subjective tinnitus much better now and also introduced the new definition of "tinnitus disorder" as suggested by DeRidder et al., last year.

The introduction was significantly improved, although I still miss the introduction of primary and secondary tinnitus cases.

it has a significant impact on everyday functioning. This is particularly important as the questionnaires generally applied in everyday clinical practice (e.g., Tinnitus Handicap Inventory) can detect the consequences regarding everyday tasks. Please include and cite.

Thank you for this addition. We have added a sentence to the Introduction and cited the work accordingly.

I have checked the manuscript many times, but unfortunately, I cannot find the sentence or the recommended references. Therefore, I include them once again. 

[Mancini PC, Tyler RS, Smith S, Ji H, Perreau A, Mohr AM. Tinnitus: How Partners Can Help? Am J Audiol. 2019 Mar 15;28(1):85-94. doi: 10.1044/2018_AJA-18-0046.]

[Mavrogeni P, Maihoub S, Tamás L, Molnár A. Tinnitus characteristics and associated variables on Tinnitus Handicap Inventory among a Hungarian population. J Otol. 2022, https://doi.org/10.1016/j.joto.2022.04.003.]

it is always a problem that ‘monitoring’ the symptoms can worsen the actual symptoms reported by the patients. This is generally a limitation of self-reported questionnaires, especially when patients have to fill in the questionnaires multiple times a day. I think this should be included as a potential limitation.

Thank you for bringing this point up. According to our own analyses, when patients consciously deal with their tinnitus daily, this does not increase their tinnitus perception as a whole. (Schlee et al (2016): “Measuring the moment-to-moment variability of tinnitus: the TrackYourTinnitus smart phone app”).

Including that reference, I find this a good discussion now; thank you for it. 

These questions can be found in Table 1, except for Question Three. Please, specify the reason why this question cannot be found there.

Question 3 (How stressful is the tinnitus right now?) is highly correlated with the targets. It would have great prediction power for the ML task at hand, but it would not give us new insights about the tinnitus disease. That’s why it has been excluded as a ML feature.

Thank you for the explanation. I think this would be interesting for the readers too; therefore, I recommend including it in the manuscript. 

Line 234. Please use the term Cramér’s V.

Thank you for the corrections. However, I have found one place where ‘Cramer’s’ can still be found (see line 245). 

Cramér’s V is not defined in range but as category (or nominal values); please correct this. There is a misunderstanding here. Cramér's V can have values from 0 to 1 (i.e., is defined in range(0,1)) and is applied in the case of categorical variables. We have changed the wording here to prevent further misunderstanding. We also included an interpretation hint for the numbers.

In the current form, the explanation is more accurate, thank you. 

Line 583. The temperature dataset. Are there any previous research data accessible on this question? Currently, only Wikipedia is referenced, and the results are not contrasted to previous research outcomes.

We have looked extensively for good sources of country temperature data and found them. We are not aware of any other publicly available temperature country datasets. We did not contrast our results with papers because there is nothing so far - to our best knowledge - that relates tinnitus to temperature. There are papers that correlate season and tinnitus, and we have described these in the Discussion and Related Work sections.

In this case, please discuss this (e.g., the correlation between tinnitus and temperature was previously not studied).

Author Response

Thank you again for your hints and comments. Please find our answers in the document attached.
